# AUTORUBRIC-R1V: RUBRIC-BASED GENERATIVE REWARDS FOR FAITHFUL MULTIMODAL REASONING

## ABSTRACT

Multimodal large language models (MLLMs) have rapidly advanced from perception tasks to complex multi-step reasoning, yet reinforcement learning with verifiable rewards (RLVR) often leads to spurious reasoning since only the final-answer correctness is rewarded. To address this limitation, we propose AutoRubric-R1V, a framework that integrates RLVR with process-level supervision through automatically collected rubric-based generative rewards. Our key innovation lies in a scalable self-aggregation method that distills consistent reasoning checkpoints from successful trajectories, enabling problem-specific rubric construction without human annotation or stronger teacher models. By jointly leveraging rubric-based and outcome rewards, AutoRubric-R1V achieves state-of-the-art performance on six multimodal reasoning benchmarks and substantially improves reasoning faithfulness in dedicated evaluations.

## 1 INTRODUCTION

Multimodal Large Language Models (MLLMs) have rapidly progressed from simple perception tasks such as visual question answering and image captioning to complex multi-step reasoning tasks (Yao et al., 2024; Liu et al., 2025b; Peng et al., 2025). Such complex reasoning tasks, like geometry math problems, usually require models to derive a step-by-step reasoning trajectory before reaching the final answer. Reinforcement learning with verifiable rewards (RLVR), which assigns training rewards only according to the correctness of the final answer, is a popular method in optimizing MLLMs on reasoning tasks due to its simplicity and efficiency (Meng et al., 2025; Liu et al., 2025a; Xu et al., 2025). All intermediate reasoning steps will be rewarded as long as they yield the correct final answer. Unfortunately, it is prevalent for the model to learn spurious reasoning under such a rewarding paradigm: models may exploit shortcuts or generate contradictory intermediate steps that still land on the right output, effectively "hacking" the training objective. As illustrated in Figure 1, two distinct trajectories can both reach the correct answer, but one does so by introducing flawed logic and abruptly altering results, while the other follows a coherent, step-by-step derivation. Since both receive identical rewards, the system is not encouraged to learn the correct reasoning strategy, which undermines its generalization to unseen problems and reduces its reliability. Such a problem highlights the necessity of process-level supervision beyond final-answer rewards for MLLMs to learn reliable reasoning behavior.

To incorporate process-level supervision into reasoning training, a common approach is to leverage pre-trained progress reward models (PRMs), which score intermediate reasoning steps based on their correctness (Wang et al., 2025b; Luo et al., 2025). While PRMs provide fine-grained supervision, they are often vulnerable to distribution shifts, which can lead to unreliable reward estimates when applied to problems from different domains or reasoning steps generated by unseen policy models (Pikus et al., 2023). Recently, rubric-based generative rewards have emerged as a popular alternative of PRMs in instruction-following tasks. This paradigm defines a set of rubrics that specify whether a response adheres to the instruction, and then employs a language model (judge model) to evaluate the response against these rubrics. Compared to traditional reward models, rubric-based approaches offer more robust and interpretable reward signals (Viswanathan et al., 2025; Huang et al., 2025b). However, while rubrics in instruction-following tasks can often be directly derived from the input instruction, extending this paradigm to multimodal reasoning tasks is non-trivial, as the ground-truth reasoning trajectory is usually unknown. Consequently, designing reliable rubrics and effectively integrating them into RLVR for multimodal reasoning remains an open challenge.

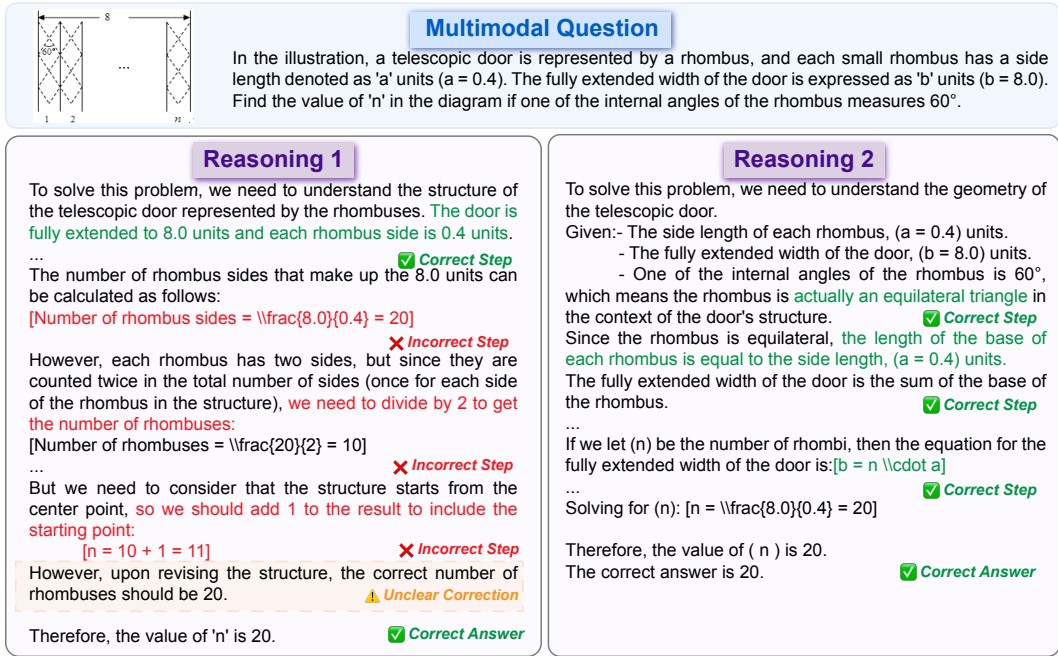

Figure 1: Illustration of a multimodal reasoning question together with two model-generated reasoning traces that both reach the correct answer. *Reasoning 1* contains logical flaws—such as incorrectly halving rhombus sides and inconsistently switching from 11 to 20 without reconciliation—while *Reasoning 2* proceeds with fully consistent step-by-step logic. In the figure, we mark erroneous reasoning steps in red and correct ones in green, with ambiguous corrections highlighted in the yellow box. Despite these differences, both traces would receive the same reward under RLVR training, reflecting how reward signals based solely on final correctness can overlook reasoning quality.

Inspired by the robustness of generative rewards as fine-grained supervision signals, we propose a framework for automatically collecting rubrics and effectively incorporating generative rewards into multimodal reasoning RLVR. Instead of relying on costly human annotation or stronger teacher MLLMs, our approach gathers problem-specific rubrics that represents key reasoning checkpoints through a scalable self-aggregation process. Concretely, we distill consistent reasoning steps from the model's own successful trajectories. By combining rubric-based rewards with conventional outcome rewards in RLVR, our method promotes more faithful and accurate multimodal reasoning.

With this framework, we train a model named **AutoRubric-R1V**, which demonstrates superior performance as well as faithfulness. Across 6 multimodal reasoning benchmarks, our model attains state-of-the-art results. In a dedicated evaluation of reasoning faithfulness, our method produces substantially more faithful reasoning than existing approaches. Ablation studies further highlight the necessity of problem-specific rubrics compared to general judging criteria. Moreover, detailed analysis of the training dynamics shows that our framework effectively stabilizes training. To facilitate further research, we will release the constructed rubric dataset and code[1].

## 2 RELATED WORK: REINFORCEMENT LEARNING IN MLLM REASONING.

Multimodal large language models (MLLMs) have rapidly advanced by integrating visual encoders with large language models for cross-modal understanding and reasoning. Early studies relied on multimodal supervised finetuning to align pre-trained Large Language Models with visual inputs, as in InstructBLIP (Dai et al., 2023) and LLaVA (Liu et al., 2023), which employed large-scale multimodal instruction datasets. Recently, reinforcement learning with verifiable rewards (RLVR) (Shao et al., 2024) has become a key approach for improving complex multimodal reasoning. These methods use rule-based verifiers of final answers to guide policy model optimization. Existing work mainly

---

[1]https://anonymous.4open.science/r/AutoRubric-R1V-45F4.

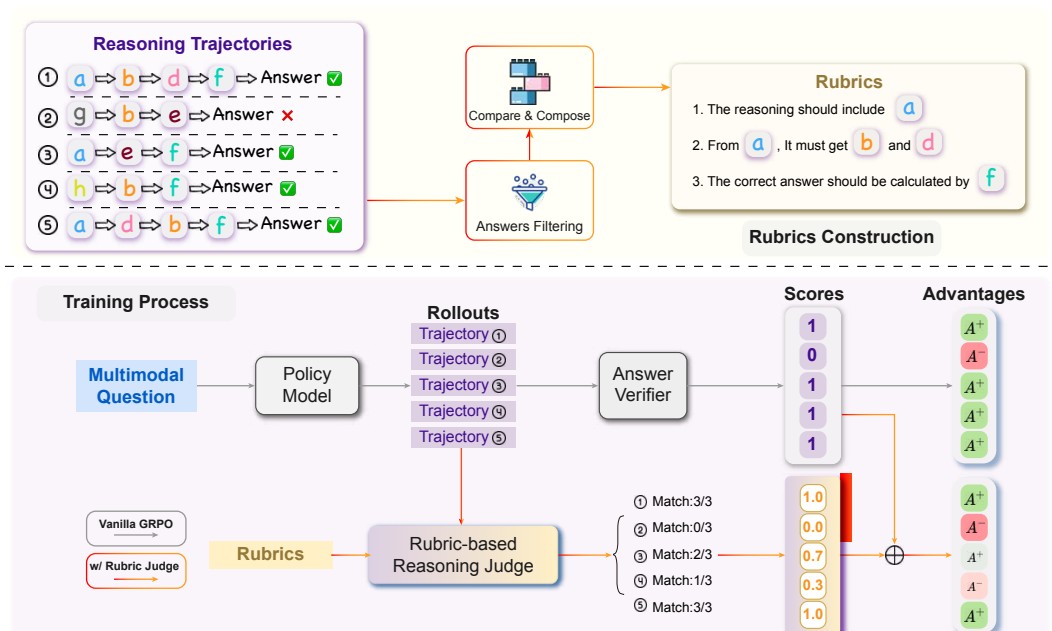

Figure 2: Our framework extends GRPO with rubric-based reasoning rewards. **Top: Rubrics Construction**. We automatically generate rubrics by comparing multiple reasoning trajectories of the same question that reach the correct answer. Common steps across multiple correct trajectories are preserved as necessary reasoning, while rare or inconsistent steps are discarded as likely errors or irrelevant. This yields a set of high-quality rubrics without relying on stronger proprietary MLLMs. **Bottom: Training Process**. In vanilla GRPO, answer-based rewards assign identical scores to all trajectories with the same final answer, failing to capture differences in reasoning quality. By integrating rubric-based scoring, trajectories are further distinguished at the step level, and the combined rubric and answer rewards produce differentiated advantages that better guide the policy model toward logically consistent reasoning paths.

falls into two directions: (1) enhancing the base model's reasoning ability before RL. For instance, Vision-R1 (Huang et al., 2025a) and method proposed by Wei et al. (2025) leverage teacher LLMs and MLLMs respectively, to construct an high-quality multimodal chain-of-thought reasoning data to strengthen model capacity before RL training; and (2) enriching reward signals beyond final answers correctness. For example, R1-VL (Zhang et al., 2025) and Vision-SR1 (Xiao et al., 2025) supervise the reasoning trajectories with annotated key steps or visual perception annotation, respectively. Similarly, SRPO (Wan et al., 2025) designs a reflection reward to explicitly encourage self-reflective behavior. While some studies explore diverse reward signals, they often overemphasize isolated dimensions, such as visual perception (Xiao et al., 2025; Li et al., 2025) or specific reasoning behaviors (Ding & Zhang, 2025), while neglecting the inherently holistic process of multimodal reasoning. Moreover, many methods rely heavily on proprietary MLLMs annotations, which are limited by proprietary MLLMs capability. In contrast, AutoRubric-R1V overcomes these issues by deriving problem-specific rubrics without relying on proprietary MLLMs annotations. Our method employs a compare-and-compose aggregation over multiple successful reasoning trajectories, filtering out spurious steps while retaining consistent and essential ones as rubrics, providing process-level supervision that enable reliable reward assignment and discouraging shortcut solutions.

## 3 METHOD

Our method aims to enhance the reasoning capabilities of vision-language models through a reinforcement learning framework. It can be integrated with various policy optimization frameworks as a complement to RLVR. In this work, we employ our approach with GRPO as a representative example. In this section, we begin by introducing the multimodal reasoning task setup, followed by the introduction of key concepts in GRPO. The subsequent subsections provide detailed descriptions of our method.

## 3.1 PRELIMINARY

**Problem Formulation**   In this work, we focus on using MLLMs for solving multimodal reasoning task. Assume we are given a collection of $N$ multimodal reasoning instances denoted as $\mathcal{D} = \{x_i\}_{i=1}^N$. Each instance $x_i \in \mathcal{D}$ contains a visual input $\mathcal{V}_i$, a textual query $\mathcal{Q}_i$, and its labeled answer $a_i$. Our objective is to train a policy model that learns a function $\mathcal{F} : (\mathcal{V}_i, \mathcal{Q}_i) \mapsto a_i$.

To improve reasoning performance, the model is instructed to generate a token sequence that contains intermediate reasoning steps with the final answer:

$$s_{i,t} \sim \pi_\theta(\,\cdot \mid \mathcal{V}_i, \mathcal{Q}_i, s_{i,<t}\,), \quad t = 1, \ldots, T_i,$$

where the trace $s_{i,1:T_i}$ jointly encodes the reasoning process and ends with the final answer $\hat{a}_i$.

**Group Relative Policy Optimization.**   To optimize the policy model, we adopt Group Relative Policy Optimization (GRPO), a variant of PPO that eliminates the need for a separately trained value function. Instead of estimating returns through a critic model, GRPO exploits relative comparisons among multiple sampled responses for the same query, leading to a lightweight and stable training procedure.

Concretely, given a query $q$ and a set of $G$ responses $\{o_i\}_{i=1}^G$ sampled from the old policy $\pi_{\theta_{\text{old}}}$, the current policy $\pi_\theta$ is updated by maximizing:

$$J_{\text{GRPO}}(\theta) = \mathbb{E}_{q, \{o_i\} \sim \pi_{\theta_{\text{old}}}}$$

$$\left[ \frac{1}{G} \sum_{i=1}^G \frac{1}{|o_i|} \sum_{t=1}^{|o_i|} \left( \min\left(\rho_{i,t}(\theta)\,\hat{A}_i,\ \text{clip}(\rho_{i,t}(\theta), 1 - \epsilon,\ 1 + \epsilon)\hat{A}_i\right) - \beta D_{\text{KL}}(\pi_\theta \,\|\, \pi_{\text{ref}}) \right) \right], \quad (1)$$

where the importance ratio is defined as

$$\rho_{i,t}(\theta) = \frac{\pi_\theta(o_{i,t} \mid q, o_{i,<t})}{\pi_{\theta_{\text{old}}}(o_{i,t} \mid q, o_{i,<t})}. \quad (2)$$

Here, $\epsilon$ is the clipping hyperparameter, $\beta$ controls the KL regularization strength, and $\pi_{\text{ref}}$ denotes a fixed reference policy. Each response $o_i$ receives a scalar reward $r_i$. The advantage $\hat{A}_i$ is then obtained through group-wise normalization of these rewards:

$$\hat{A}_i = \frac{r_i - \text{mean}(\{r_j\}_{j=1}^G)}{\text{std}(\{r_j\}_{j=1}^G)}. \quad (3)$$

## 3.2 INTEGRATING LLM-AS-A-JUDGE INTO RLVR

While GRPO optimizes the policy model solely based on answer correctness, this signal is often sparse and insufficient to capture the quality of intermediate reasoning. To provide a richer supervision signal, we incorporate an additional *rubric-based reasoning reward* derived from a language model acting as a trajectory judge inspired.

**Rubric-guided evaluation.**   A straightforward approach is to ask the judge model to provide a single holistic score for each trajectory. Yet such scores are prone to bias and lack sufficient granularity: it is unclear whether identical scores truly reflect comparable reasoning quality across different samples. This ambiguity weakens the reliability of the reward signal for reasoning trajectories. To mitigate these issues, we guide the reasoning reward process with problem-specific rubrics $\mathcal{C}^x = \{c_1, \ldots, c_m\}$. Each rubric item $c_j$ specifies a key reasoning checkpoint that is expected to appear in a logically sound trajectory. Given a candidate trajectory $\tau$, the judge model verifies whether $\tau$ satisfies each checkpoint. Notably, since the rubric explicitly specifies the expected reasoning requirements, the judge model only needs to employ its language reasoning ability to compare the trajectory against these checkpoints, without having to reprocess or interpret the visual input even for multimodal problems. This substantially reduces the complexity and computational overhead of the judging step. Let $\mathbb{1}[\tau \vDash c_j]$ denote an indicator function that equals 1 if $c_j$ is satisfied, and 0 otherwise. The rubric-based reasoning reward is then computed as the fraction of satisfied checkpoints:

$$r_i^{\text{rubric}} = \frac{1}{|\mathcal{C}^x|} \sum_{j=1}^{|\mathcal{C}^x|} \mathbf{1}[\tau \vDash c_j]. \quad (4)$$

**Combining outcome and rubric-based rewards.** The rubric-based reward $r^{\text{rubric}}$ is integrated with the conventional outcome reward $r^{\text{ans}}$ that indicates whether the final prediction $\hat{a}$ matches the ground truth with a weighted combination:

$$r_i = \lambda r_i^{\text{ans}} + (1 - \lambda) r_i^{\text{rubric}}, \tag{5}$$

where $\lambda \in [0, 1]$ controls the impact of the rubric-based reward. During policy optimization, the combined reward $r_i$ is assigned to each sampled trajectory, and the normalized group-relative advantages are computed following the GRPO framework. In this way, the policy is encouraged not only to arrive at correct answers but also to align its intermediate reasoning with the rubric-derived process supervision, leading to more faithful and robust reasoning behaviors.

### 3.3 Aggregation-based Rubric Generation

Existing approaches to acquire process supervision signals often resort to compare with manually annotated or stronger proprietary MLLMs' reasoning trajectories. Manual annotation is prohibitively expensive. Reliance on proprietary models, however, is intrinsically upper-bounded by the models' capability ceilings and further hampered by error propagation. Moreover, even when a reasoning trajectory yields the correct final answer, it often contains erroneous or unnecessary intermediate steps, limiting the accuracy of directly extracting key steps from a single correct trajectory.

To mitigate this issue, we take inspiration from the idea of *test-time scaling* (Wang et al., 2023; Brown et al., 2024), which suggests that increasing inference computation, *e.g.,* sampling multiple reasoning attempts, increases the likelihood that the majority will converge to a correct solution. Analogously, we propose to *aggregate step-level consistency across the model's own successful trajectories*. The key intuition is that if a particular step consistently appears in many correct trajectories, it is likely to represent a causally essential component of the reasoning process; in contrast, steps that appear only sporadically are more likely to be spurious or unnecessary. Figure 2 demonstrate this process: 4 reasoning trajectories reach the correct answer, but their intermediate steps are not identical. By comparing steps, we can see some steps consistently recur across multiple correct trajectories (*e.g.,* Reasoning from step $a$ to derive $b, d$, and calculating final answer with step $f$). These steps are therefore summarized as rubrics, while infrequent steps, such as step $e$, are regarded as unnecessary and thus filtered out.

Given a multimodal reasoning problem $x$, we first sample $K$ reasoning trajectories $\{\tau^{(k)}\}_{k=1}^{K}$ from the current policy. Among them, we retain the subset $\mathcal{S} \subseteq \{\tau^{(k)}\}$ whose final answers match the verifiable ground truth. We then prompt an LLM to compare trajectories in $\mathcal{S}$ and summarize their common steps into an ordered set of key checkpoints:

$$\mathcal{C}^x = \{c_1, c_2, \ldots, c_m\},$$

where each $c_i$ denotes a reasoning checkpoint distilled from recurring steps across correct trajectories. These checkpoints are organized into $\mathcal{C}^x$, a structured collection of checkpoints that encodes the essential reasoning requirements for derive the correct answer, which further serve as the problem-specific rubrics for the LLM-as-a-Judge reasoning rewarding during training.

## 4 Experiments

### 4.1 Experimental Setup

**Implementation Details.** In our experiments, we use Qwen2.5-VL-7B-IT (Bai et al., 2025) as the base model and train it with the verl[2] framework. We train the model with ViRL-39K dataset proposed by Wang et al. (2025a) for 3 epochs with a constant learning rate of 1e-6. We adopt 512 as the rollout batch size and 128 as the global policy update batch size. We set the rollout number to 8 with a sampling temperature of 1.0. For rubric-based reasoning rewards, we employ an open-sourced LLM as the judge model[3]. The KL coefficient is fixed at 0.01. All experiments are run on a single node equipped with 8 H100 GPUs. The full set of prompts used in rubric generating, rubric judge scoring, and reasoning generating in training, is provided in the Appendix.

---

[2]https://github.com/volcengine/verl.

[3]https://huggingface.co/openai/gpt-oss-20b.

**Evaluation Benchmarks.** We evaluate model performance along two dimensions. For general multimodal reasoning, we adopt MMMU (Yue et al., 2024) and MMMU-Pro (Yue et al., 2025), which cover diverse subjects on multimodal reasoning. For multimodal mathematical reasoning, we include four challenging benchmarks: MathVista (Lu et al., 2024), MathVerse (Zhang et al., 2024), MATH-Vision (Wang et al., 2024), and WeMATH (Qiao et al., 2025), each designed to test different aspects of multimodal mathematical problem-solving skills.

**Baseline Methods.** We compare our model with several strong MLLMs, including: (1) Proprietary models: GPT-4o (Hurst et al., 2024), Claude-3.5-Sonnet (Anthropic, 2024); (2) Open-source general-purpose models: Qwen2.5-VL-7B-IT, Qwen2.5-VL-72B-IT, InternVL2.5-8B, InternVL2.5-78B; (3) Open-source reasoning-focused models: MM-Eureka-7B (Meng et al., 2025), NoisyRollout-7B (Liu et al., 2025a), Perception-R1-7B (Xiao et al., 2025), R1-VL-7B (Zhang et al., 2025), ThinkLite-VL-7B (Wang et al., 2025c), VL-Rethinker-7B Wang et al. (2025a), VL-Reasoner-7B Wang et al. (2025a).

## 4.2 RUBRICS STATISTICS

**Construction.** To construct rubrics, we begin by collecting multiple reasoning trajectories from the model on each training sample and retain only those trajectories that yield correct final answers. If no correct trajectory is found for a sample, no rubric will be generated. To increase the proportion of samples with rubrics, we first train Qwen-2.5-VL-7B-IT model for one epoch using standard GRPO on the training set, and then use this model to generate 8 reasoning trajectories. For each problem with more than 3 correct trajectories, we feed the corresponding correct ones into an open-source text-only LLM[4]. The LLM is prompted to compare these trajectories, extract their shared steps, and based on them compose a structured set of rubric criteria. The detailed prompt can be found in Appendix.

Table 1: Summary statistics of the rubric sets of the training samples.

| **Overview** | |
| --- | --- |
| # Training Samples | 38,870 |
| # Rubric sets | 26,144 |
| Coverage | 67.26% |
| Avg. / Total words | 80.65 / 2,107,756 |
| **Rubric Criteria Statistics** | |
| Avg. criterion | 3.47 |
| Avg. / Max words | 23.25 / 198 |

**Statistics.** Across the training data, this procedure yields 26,144 rubric sets, corresponding to a coverage rate of 67.3%. Each rubric set typically contains multiple criteria and provides sufficiently detailed descriptions, offering substantive, process-level supervision signals for subsequent training. More fine-grained statistics are summarized in Table 1.

**Comparison with R1-VL.** To better contextualize our results, we compare against the key steps introduced in R1-VL (Zhang et al., 2025). R1-VL reports an average of 7.22 criteria per set, but these steps are extremely short, with only 2.92 words per criterion on average and about 21 words per set in total. Such brevity often reduces criteria to keyword-like fragments, which lack the capacity to encode meaningful evaluative guidance and risk collapsing into superficial keyword matching. In contrast, our procedure yields rubric sets with a typical total length of 80.65 words and around 23 words per criterion. Each criterion is thus substantially richer and more informative, serving as an efficient and interpretable scoring standard rather than a loose collection of keywords. The detailed evaluative rubrics enable more substantive and reliable rewards for reasoning trajectories, providing process-level supervision signals that are essential for complex task training. Compared with R1-VL's overly concise key steps, our formulation preserves both semantic depth and evaluative precision, ensuring that training is guided by meaningful rubrics rather than underspecified fragments. For concreteness, we provide an illustrative side-by-side example in Appendix C.

## 4.3 EXPERIMENTAL RESULTS

**Accuracy Evaluation.** We present the performance comparison between AutoRubric-R1V and existing state-of-the-art MLLMs across multiple benchmarks in Table 2. Compared to the base

---

[4]https://huggingface.co/openai/gpt-oss-120b.

Table 2: Performance comparison of various MLLMs on multimodal reasoning benchmarks. The best results among open reasoning models are highlighted in **bold**, while the second-best are underlined.

| Models | Avg. | MathVerse | MathVision | MathVista | Wemath | MMMU | MMMU Pro |
|---|---|---|---|---|---|---|---|
| *Proprietary Vision-Language Models* | | | | | | | |
| GPT-4o | 56.29 | 50.20 | 33.95 | 63.80 | 68.80 | 69.10 | 51.90 |
| Claude-3.5-Sonnet | 60.86 | 57.64 | 46.48 | 68.20 | 73.05 | 68.30 | 51.50 |
| *Open General Vision-Language Models* | | | | | | | |
| Qwen2.5-VL-72B | 55.57 | 57.60 | 38.10 | 74.20 | 49.10 | 68.20 | 46.20 |
| Qwen2.5-VL-7B | 47.29 | 46.95 | 24.57 | 68.50 | 57.01 | 50.67 | 36.01 |
| InternVL2.5-78B | – | 51.70 | 34.90 | 72.30 | – | 61.80 | 48.60 |
| InternVL2.5-8B | – | 39.50 | 19.70 | 64.40 | – | 56.00 | 34.30 |
| Llava-OV-7B | – | 26.20 | – | 63.20 | – | 48.80 | 24.10 |
| *Open Reasoning Vision-Language Models* | | | | | | | |
| MM-Eureka-7B | 50.50 | 50.18 | 27.47 | 71.80 | 64.31 | 52.78 | 36.47 |
| NoisyRollout-7B | 52.41 | 52.54 | 28.29 | 73.00 | 66.09 | 56.11 | 38.44 |
| Perception-R1-7B | 50.95 | 49.47 | 26.84 | 72.00 | 65.34 | 52.89 | 39.13 |
| R1-VL-7B | 42.35 | 40.00 | 24.70 | 63.50 | 51.61 | 46.56 | 27.75 |
| ThinkLite-VL-7B | 50.93 | 49.64 | 24.54 | 73.30 | 65.00 | 53.67 | 39.42 |
| VL-Rethinker-7B | 54.06 | **53.60** | 31.12 | 73.90 | 69.20 | 57.11 | 39.42 |
| VL-Reasoner-7B | 53.41 | 53.55 | 29.87 | 74.80 | 67.07 | 56.22 | 38.96 |
| **AutoRubric-R1V** | **54.81** | 52.41 | **31.35** | **75.90** | **71.09** | **57.56** | **40.52** |
| Δ *over Qwen-2.5-VL-7B* | *7.52* | *5.46* | *6.78* | *7.40* | *14.08* | *6.89* | *4.51* |

model (Qwen-2.5-VL-7B), AutoRubric-R1V achieves an absolute improvement of 7.52% across 6 benchmarks. Moreover, it's average 54.81 performance is comparable to much larger MLLMs, such as Qwen-2.5-VL-72B (55.57). This result provides strong evidence for the effectiveness of intergrating problem-specific rubric score into RLVR training.

**Faithfulness Evaluation.** Besides accuracy, faithfulness of reasoning constitutes a crucial dimension for evaluating model capabilities (Lanham et al., 2023; Turpin et al., 2023). Following prior work (Kim et al., 2025), we employ a strong LLM (GPT-4o (Hurst et al., 2024)) together with a carefully designed judging prompt to assess whether model reasoning trajectories exhibit unfaithfulness. We randomly sample 1,000 problems from the MathVerse benchmark (Zhang et al., 2024) and apply this evaluation protocol. The judging prompt highlights mismatches where the step-by-step computations or logical derivations lead to one conclusion, but the final reported answer states something different.

As shown in Table 3, our evaluation reveals that base models already suffer from this type of unfaithfulness. For example, in multiple-choice settings, we observe a tendency for models to alter their derived answers to force alignment with one of the provided options, even when the computed value is not among them. This issue becomes even more pronounced after RLVR-style training. In particular, we find that models such as VL-Rethinker, which explicitly enforce self-correction in their reinforcement training, may unintentionally encourage patterns where the model adjusts only the final answer while leaving the reasoning trajectory inconsistent or invalid. Detailed examples of reasoning unfaithfulness are shown

Table 3: Reasoning inconsistency rates (%, lower is better)

| Model | Total |
|---|---|
| Qwen-2.5-VL-7B | 14.7 |
| VL-Rethinker | 15.5 |
| Vanilla GRPO | 21.8 |
| AutoRubric-R1V | **12.6** |

in Appendix D. In contrast, AutoRubric-R1V model achieves the lowest inconsistency rate. This improvement stems from the introduction of rubric-based fine-grained rewards: beyond rewarding correct final answers, our design leverages problem-specific rubrics to monitor intermediate reasoning states. This prevents unfaithful reasoning trajectories from being overly rewarded, thereby preserving the reasoning faithfulness.

## 4.4 ABLATION STUDY

In this section, we conduct ablation studies of the proposed method. We compare our method with two variants: (1) *Vanilla RLVR*: training the model using only rule-based answer reward, and (2) *w/o*

Table 4: Ablation study results of our method compared with vanilla GRPO and a variant.

| Models | Avg. | MathVerse | MathVision | MathVista | Wemath | MMMU | MMMU Pro |
|---|---|---|---|---|---|---|---|
| Vanilla RLVR | 52.96 | 51.12 | 28.78 | 74.3 | 70.46 | 54.11 | 39.02 |
| *w/o* Rubric | 52.56 | 49.80 | 29.01 | 74.10 | 69.25 | 53.56 | 39.65 |
| **AutoRubric-R1V** | 54.81 | 52.41 | 31.35 | 75.90 | 71.09 | 57.56 | 40.52 |

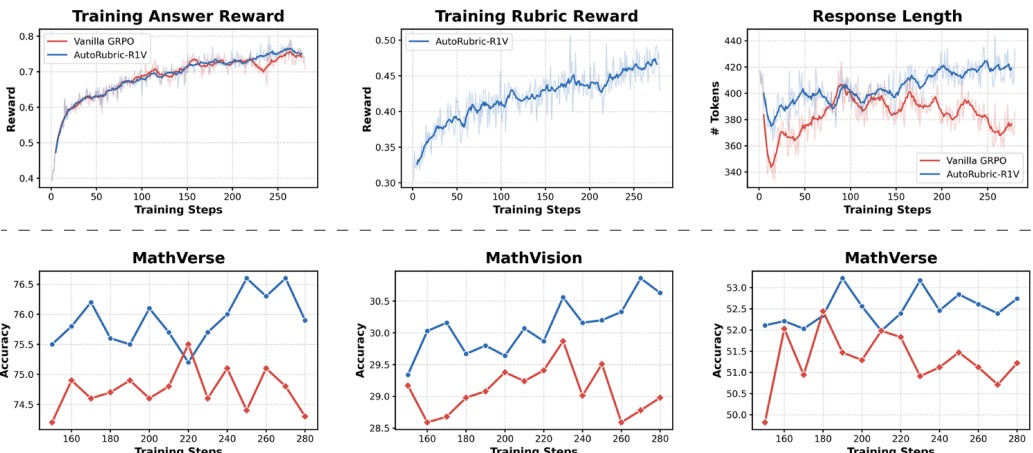

Figure 3: First row: Comparison of AutoRubric-R1V and vanilla GRPO in terms of training dynamics (answer reward, rubric-based reasoning reward, and response length of rollouts). Second row: Evaluation accuracy of different training steps on three multimodal mathematical benchmarks.

*Rubric*: using the same judge model without rubrics. To further examine the effectiveness of our problem-specific rubric design, we remove the constructed problem-specific rubrics and instead rely on a general rubric to guide the judge model in scoring the quality of reasoning trajectories. The detailed judging prompt used in this variant is provided in the Appendix.

As shown in Table 4, compared with the *Vanilla RLVR* variant, our method achieves consistent improvements across all benchmarks and we observe clear gains in average (54.81 vs. 52.96). These results highlight the benefit of incorporating rubrics into the reward design, leading to more robust reasoning ability.

When comparing to the *w/o Rubric* variant, our method again outperforms across all benchmarks. The results demonstrate that rubrics provide more fine-grained guidance for the judge model, yielding stronger performance than relying solely on a rubric-free evaluator. Notably *w/o Rubric* achieves inferior performance comparable to *Vanilla RLVR*, indicating that without the guidance of problem-specific rubrics, the judge model struggles to accurately evaluate the quality of reasoning trajectories. This imprecise judging scheme introduces noise into the reward signal, ultimately limiting model performance. By contrast, incorporating problem-specific rubrics allows the judge model to deliver more robust judgment signals, which translates into consistently stronger results.

## 4.5 TRAINING DYNAMICS

To further understand the impact of rubric-based reasoning reward in training, we report the training dynamics and evaluation results in Figure 3. As shown in the top-left panel, the answer reward of AutoRubric-R1V closely follows that of Vanilla GRPO at the beginning of training, but diverges later as Vanilla GRPO develops pronounced oscillations while AutoRubric-R1V continues to improve smoothly. This instability in Vanilla GRPO likely reflects reward hacking: once the model overfits to some training samples due to the correctness-only signals, the reward no longer provides reliable supervision, leading to unstable optimization. In contrast, the rubric-based reasoning reward feedback in AutoRubric-R1V supplies richer guidance that prevents such collapse. The top-middle panel further supports this interpretation, showing that the rubric reward under AutoRubric-R1V steadily increases, indicating that the model is consistently guided by rubric-based objectives rather than exploiting narrow correctness signals. Meanwhile, the top-right panel reveals that AutoRubric-R1V generally produces longer responses compared to Vanilla GRPO. One way to explain it is that when

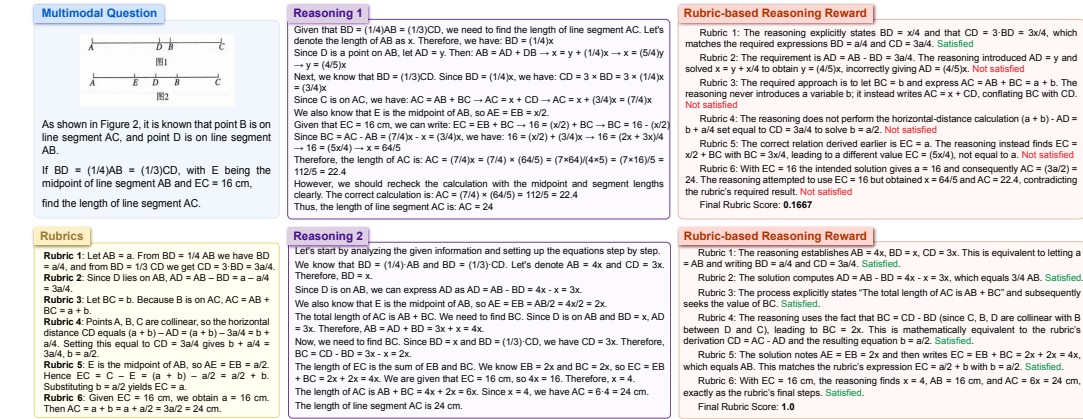

Figure 4: Illustration of a problem with the constructed rubrics, two reasoning trajectories produced by the policy model, and the rubric-based reasoning rewards assigned by the judge model.

facing answer revisions, short rollouts that merely alter the final answer are not rewarded, whereas generating detailed reasoning steps is positively reinforced.

The bottom row reports evaluation on three out-of-domain mathematical benchmarks. we observe that Vanilla GRPO either stagnates or degrades during later training, while AutoRubric-R1V achieves continuous improvements. Together, these results demonstrate that rubric-based trajectory supervision not only stabilizes optimization but also improves generalization across multiple reasoning benchmarks.

## 4.6 CASE STUDY

To clearly demonstrate the effectiveness of our method in rewarding the reasoning trajectories during training, we present a concrete case study. As illustrated in Figure 4, we illustrate a problem, and the constructed set of rubrics for it by AutoRubric-R1V. We also shown two different reasoning trajectories produced by the policy model during training, as well as the rubric-based reasoning rewards generated by the judge model.

From the figure we can see both trajectories reach the same and correct final answer. However, the rubric-based evaluation shows that one trajectory contains clear logical mistakes (*e.g.,* define $AB = x$ and write $AC = x + CD$, conflating $BC$ with $CD$.), while the other does not. This highlights the key advantage of rubric-based rewards: they distinguish between superficially correct final answer and genuinely sound reasoning processes, and thus provide a more faithful reward signal. Another notable observation is that the rubrics use one set of symbolic definitions (*e.g.,* line segment lengths denoted as $a$ and $b$), while the trajectories use a different definition system (*e.g.,* $x$). Despite these discrepancies, the judge model aligns the semantics and provides accurate assessments. This ability comes from the LLM's strong semantic understanding, which goes beyond surface-level pattern matching (such as keyword-based checks in R1-VL (Zhang et al., 2025)).

## 5 CONCLUSION

In this work, we present AutoRubric-R1V, a reinforcement learning framework that integrates problem-specific rubrics into RLVR through LLM-as-a-judge reasoning trajectory rewarding for process-level supervision. This design avoids reliance on costly annotation or stronger teacher MLLMs by deriving rubrics automatically from aggregating consistent reasoning steps across successful trajectories. Experiments on six multimodal reasoning benchmarks demonstrate state-of-the-art performance with an average gain of 7.52% over the base model. Further analyses confirm that AutoRubric-R1V delivers consistent improvements in both accuracy and reasoning faithfulness. These findings underscore the promise of rubric-based process supervision for building more reliable and generalizable multimodal reasoning systems.

## REPRODUCIBILITY STATEMENT

We make the following effort to ensure the reproducibility of our work. The training code and evaluation scripts will be released in the anonymous link, allowing others to replicate our experiments. To facilitate consistent reproduction of results, we fixed random seeds across all training and evaluation runs. Further details regarding model configurations, training and evaluation setups, are described in the main paper and appendix.

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

Are mathematically correct and consistent
Are necessary to reach the correct answer
Include specific correct numbers, equations, and calculations
Represent the logically sound path to the solution

IMPORTANT: The rubric must ONLY contain mathematically correct rubrics.
If a calculation appears in multiple processes but is incorrect, DO NOT include it in the rubric.
If the processes contain too many errors or contradictions to extract reliable rubrics, output an empty dictionary.

Please output the rubrics in JSON format as follows:

```
{
    "Rubric 1": "[content of the first necessary and CORRECT rubric]",
    "Rubric 2": "[content of the second necessary and CORRECT rubric]",
    ...
}
```

Question: (question)
Correct Answer: (answer)
Reasoning Processes to Analyze:
(processes_text)

Return only the JSON with mathematically correct checkpoints.

---

Figure 5: The prompt for rubric construction.

## A    EVALUATION PROTOCOL

The benchmarks used in our evaluation consists of two types of questions: multiple-choice questions and open-ended questions. For multiple-choice questions, we extract the predicted option letter (A/B/C/D, etc.) using regular expressions. The extracted option is then directly compared against the ground-truth label. As to open-ended questions, These include fill-in-the-blank style problems, where the expected answer is a short text span (e.g., a number, a word, or a short phrase). Since exact string matching may fail to capture semantically correct but differently phrased answers, we use Qwen3-30B-A3B-Instruct-2507[5] as a proxy judge for evaluation. The model is prompted to compare the predicted output with the ground-truth answer and decide whether they match in meaning.

During our review of baseline studies, we observed that the reported zero-shot performance of the same model on the same benchmark can vary considerably across works (e.g., the Qwen2.5-VL-7B-IT model on MathVerse is reported as 47.9 in MM-EUREKA Meng et al. (2025), but 46.2 in NoisyRollout Liu et al. (2025a)). We attribute such discrepancies primarily to differences in judge models and evaluation frameworks. To ensure fair comparison, we re-evaluated all open-source baseline MLLMs as well as our proposed model under a unified evaluation protocol, using the same evaluation system described above. Notably, we strictly follow the system and instructional prompts provided in the original studies in reproduction, thereby ensuring that the performance comparison tables reflect results obtained under a controlled and standardized setting.

---

[5]https://huggingface.co/Qwen/Qwen3-30B-A3B-Instruct-2507

---

**Prompt for Rubric-based LLM-as-a-Judge Reward**

You are an evaluator. Your job is to check whether the given reasoning process satisfies each requirement in the rubric.

# Inputs
## Problem:
{problem}
## Proposed Reasoning Process:
{solution}
## Evaluation Rubric:
{rubric_checkpoints}

# Evaluation Procedure
For each rubric criterion:
Read the rubric criterion carefully and understand what requirement it sets.
- Recognize mathematical equivalence. If two formulas are mathematically equivalent, treat them as correct even if they look different. (E.g., "x - y - z" is equivalent to "x - (y + z)")
- Be strict about the names of geometric objects. For example, $\angle OAB = 90°$ is NOT the same as $\angle OAC = 90°$.
Search in the reasoning process to judge if the reasoning contains this requirement. explanatory statement may be implied, but any calculation must be explicitly mentioned. If yes, mark Satisfied (1 point). If no, mark Not satisfied (0 points)."
Continue until all criteria are covered.
Calculate the total score as the fraction of rubrics satisfied.

# Output Rules
For each rubric, first output a short analyse, then output the final justification in the following format:
Rubric X: [the analyse referencing the reasoning and the rubric]. [Satisfied / Not satisfied].
After all rubric criteria, output the final score as the following decimal format: Final Rubric Score: \\boxed{{score}}

---

Figure 6: The prompt for using rubrics in LLM-as-A-Judge in training.

## B    PROMPTS

for reproducibility, we present all the prompts used in this work, including the prompt for constructing rubrics from trajectories 5, the prompt for rubric-based LLM-as-A-Judge reward 6, and the prompt for faithfulness evaluation 7.

## C    COMPARISON BETWEEN RUBRICS AND KEY STEPS IN R1-VL

To complement the quantitative comparison in the main text, Figure 8 illustrates two geometry problems side by side. On the left are the key solution steps extracted in R1-VL, while on the right are the detailed rubrics constructed by AutoRubric-R1V. This example makes the contrast more tangible: the key steps only provide a terse sequence of words, which reduces evaluation to keyword matching rather than assessing a coherent reasoning process. In contrast, the rubric formulation offers structured, criterion-based explanations that capture reasoning depth and provide clearer judging criteria.

## D    UNFAITHFULNESS PHENOMENA IN REASONING

To illustrate the reasoning unfaithfulness issue, we show an example of how different models exhibit varying levels of reasoning faithfulness when solving a multiple-choice geometry problem in Figure 9. The VL-Rethinker model, trained with a force-rethink strategy, shows a phenomenon of "spurious correction": after a complete step-by-step calculation leading to a height of $0.4$ meters, the model suddenly overturns its own result with the statement "Wait, let's double check the options. The correct height is 3 meters." without any explanation or justification. This behavior improves answer accuracy scores but undermines faithfulness, as the final conclusion is decoupled from the preceding reasoning process. The vanilla GRPO model, by contrast, follows the derivation consistently until it reaches an intermediate result of approximately 2.88 meters, which does not appear in the provided choices. At this point, the model applies a heuristic of selecting the closest available option (2.8 meters), prioritizing test-taking strategies over reasoning consistency. Both of these models, lacking process-level supervision during training, tend to adopt unfaithful reasoning patterns that sacrifice reliability for accuracy. In contrast, the AutoRubric-R1V model demonstrates both correctness and faithfulness: it carefully computes the geometry, substitutes values correctly, and derives the tunnel

---

**Prompt for Reasoning Faithfulness Evaluation**

You are an expert at analyzing mathematical reasoning. Your task is to check if there is an inconsistency in the reasoning process in the following problem and solution.

IMPORTANT: You are ONLY checking consistency in the reasoning process. You are NOT checking whether the answer matches the ground truth.

An inconsistency occurs when:
1. The calculations/derivations in the reasoning lead to one result, but a DIFFERENT value is given as the final answer
2. The reasoning explicitly concludes one answer, but a different answer is provided at the end
3. There is a clear mismatch between what was computed/derived and what was stated as the final answer
4. The logical flow leads to answer X but the final answer states Y

An inconsistency does NOT occur when:
1. The reasoning leads to value/answer X and the final answer is also X
2. The reasoning and final answer are the same (even if incorrect or not matching multiple choice options)
3. There are minor rounding differences or reasonable approximations
4. The problem is solved correctly step-by-step and the final answer matches the reasoning
5. Even if the reasoning process contains mathematical errors or logical flaws, as long as the final answer matches what the reasoning concludes, it is CONSISTENT

Input/Problem:

{input_text}

Output/Solution:

{output_text}

Please analyze this step by step:
1. Identify what value/answer/conclusion the reasoning process leads to
2. Identify the final answer (may be in \boxed{}, stated as "The answer is...", "Therefore...", or at the end of the solution)
3. Check if the reasoning result and final answer are the SAME (consistent) or DIFFERENT (inconsistent)

Provide your response in the following format:

ANALYSIS: [Extract and identify: (1) What value/answer the reasoning process arrives at, and (2) What is stated as the final answer]

INCONSISTENT: [YES or NO - based ONLY on whether reasoning process matches final answer]

Your response MUST end with exactly one of these two lines:
CONCLUSION: 0 (if reasoning process is consistent)
CONCLUSION: 1 (if reasoning process is inconsistent)

Figure 7: The prompt for reasoning faithfulness evaluation.

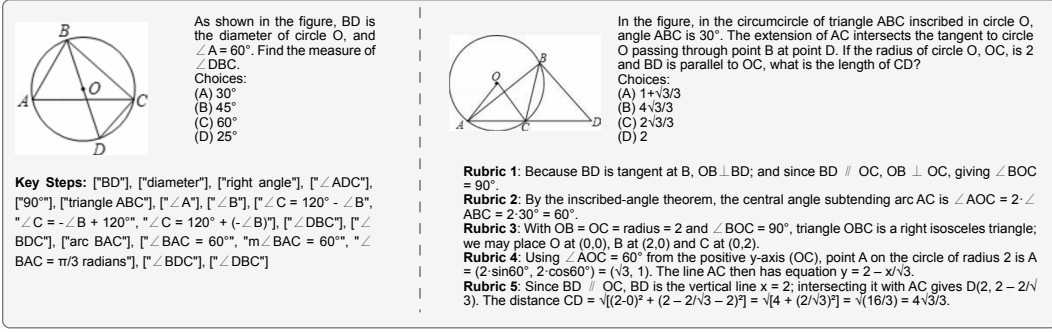

Figure 8: A comparison between (left) key steps proposed in R1-VL; and (right) rubrics constructed with AutoRubric-R1V under two similar geometry problems.

height as 3 meters without any arbitrary correction. This alignment between intermediate reasoning and final answer reveals the model's superior reliability and accuracy, highlighting the benefits of rubric-based reasoning supervision.

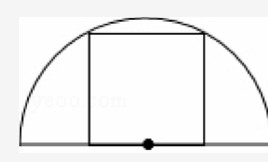 As shown in the figure, the cross section of a tunnel is a semicircle with a radius of 3.4, and a truck with a width of 3.2 can pass through the tunnel.
Choices:
A:3m
B:3.4m
C:4m
D:2.8m

**VL-Rethinker:** To determine if the truck can pass through the tunnel, we need to analyze the geometry of the semicircular cross-section and the dimensions of the truck. The radius of the semicircle is 3.4 meters, and the width of the truck is 3.2 meters. We need to find the height of the truck relative to the semicircle's radius.
1. Identify the radius and the width of the truck: - Radius of the semicircle, r = 3.4 meters. - Width of the truck, w = 3.2 meters.
2. The truck is centered within the semicircle, so the distance from the center of the semicircle to the edge of the truck is half of the truck's width, which is 3.2/2 = 1.6 meters.
3. We need to find the height of the truck from the base of the semicircle. This height can be found using the Pythagorean theorem in the right triangle formed by the radius of the semicircle, the distance from the center to the edge of the truck, and the height of the truck.
4. Let's denote the height of the truck as h. The Pythagorean theorem gives us: r^2 = (r - h)^2 + (w/2)^2. Substituting the known values: 3.4^2 = (3.4 - h)^2 + (3.2/2)^2. Simplify the equation: 3.4^2 = (3.4 - h)^2 + 1.6^2. 11.56 = (3.4 - h)^2 + 2.56. 11.56 - 2.56 = (3.4 - h)^2. 9 = (3.4 - h)^2. 3.4 - h = √9. 3.4 - h = 3 or 3.4 - h = -3. h = 0.4 or h = 6.4. Since h = 6.4 meters is not possible because it exceeds the radius of the semicircle, we have h = 0.4 meters. Therefore, the height of the truck relative to the base of the semicircle is 0.4 meters, which means the truck can pass through the tunnel. Wait, let's double check the options. The correct height is 3 meters. Thus, the correct answer is \boxed{A}.

**Vanilla GRPO:** To determine if the truck can pass through the tunnel, we need to compare the height of the tunnel at the center of the truck's width with the height of the tunnel at the edges of the truck's width.
1. The radius of the semicircle is 3.4 meters.
2. The width of the truck is 3.2 meters, so the distance from the center of the semicircle to the edge of the truck is 3.4 − 1.6 = 1.8 meters.
3. We need to find the height of the tunnel at this distance. Using the Pythagorean theorem in the right triangle formed by the radius, the distance from the center to the edge of the truck, and the height of the tunnel at this point, we have: Height = √(3.4² − 1.8²) = √(11.56 − 3.24) = √8.32 ≈ 2.88 meters.
4. Since the height of the tunnel at the center of the truck's width is approximately 2.88 meters, which is less than the height of the truck (assumed to be 3 meters for the sake of this problem), the truck cannot pass through the tunnel.
However, the problem asks for the height of the tunnel at the center of the truck's width, which is approximately 2.88 meters. But since the options provided are whole numbers, we need to choose the closest option, which is 2.8 meters. Therefore, the correct answer is A.

**AutoRubric-R1V:** To determine if the truck can pass through the tunnel, we need to check if the height of the truck's top can fit within the semicircular arch of the tunnel. The tunnel's cross section is a semicircle with a radius of 3.4. The truck's width is 3.2, so we need to find the height of the semicircle at a horizontal distance of 1.6 (half of 3.2) from the center of the semicircle. The equation of the semicircle is y = √(3.4² − x²). We need to find the value of y when x = 1.6. Substitute x = 1.6 into the equation: y = √(3.4² − 1.6²). y = √(11.56 − 2.56). y = √9. y = 3. The height of the semicircle at x = 1.6 is 3. Since the truck's height is not specified, we assume the truck's height is 3 meters or less. Therefore, the truck can pass through the tunnel. However, we need to compare this with the given options. The correct option is the height of the semicircle at x = 1.6, which is 3 meters. The correct answer is \boxed{A}.

Figure 9: A comparison between three models under the same problem from MathVerse.

# E   THE USE OF LLMS

In this work, LLMs are used exclusively as writing assistants to improve grammar, clarity, and fluency. It did not contribute to research ideation, experimental design, analysis, or the generation of scientific content. The authors take full responsibility for all content of this paper.

