# OpenReview forum: "AutoRubric-R1V: Rubric-Based Generative Rewards for Faithful Multimodal Reasoning"
_ICLR.cc/2026/Conference — ICLR 2026 Conference Withdrawn Submission_

### Official Review · Reviewer_ToJF · 2025-10-30

**Soundness:** 2
**Presentation:** 2
**Contribution:** 2
**Rating:** 4
**Confidence:** 4

**Summary:**

The paper proposes AutoRubric-R1V, a reinforcement learning framework that integrates rubric-based generative rewards into RL with verifiable rewards (RLVR) for multimodal reasoning. AutoRubric automatically constructs problem-specific rubrics by aggregating consistent reasoning checkpoints across multiple correct trajectories. The rubric-guided judge then scores reasoning trajectories at the process level. Experiments on six multimodal reasoning benchmarks show a 7.5-point absolute improvement over the Qwen-2.5-VL-7B baseline and reduced reasoning inconsistency.

**Strengths:**

1. Clear motivation and illustration. Figure 1 and Figure 2 effectively explain why correctness-only rewards encourage spurious reasoning.

2. Automatic rubric generation. The aggregation of common steps from successful trajectories is elegant and removes the need for costly human annotation.

**Weaknesses:**

1. **Limited conceptual novelty.** The method is largely an incremental extension of prior RLVR frameworks using GRPO, such as R1-VL and Vision-SR1, with the addition of a rubric-guided judge. However, rubric-based evaluation is not a new concept and has been explored in several previous works [1-3]. The proposed aggregation mechanism closely resembles majority-vote self-consistency or test-time scaling, making it more of a data-level heuristic than a methodological innovation.

[1] Reinforcement Learning with Rubric Anchors

[2] Breaking the Exploration Bottleneck: Rubric-Scaffolded Reinforcement Learning for General LLM Reasoning

[3] Rubrics as Rewards: Reinforcement Learning Beyond Verifiable Domains

2. **Limited empirical advantage over recent reasoning models.** While the proposed method shows clear gains over the vanilla **Qwen-VL-7B** baseline, the improvements compared with other **open-source reasoning V-L models** are small. The reported gains do not clearly demonstrate a significant step forward in reasoning capability.

3. The paper does not clarify whether experiments were conducted under the same rollout budget or token cost as baselines like R1-VL. Without controlling for computational resources, the reported improvements may partly arise from increased sampling during rubric construction rather than from the method itself.

**Questions:**

1. How sensitive is performance to the number of sampled trajectories (K) and rubric coverage rate?

2. Can rubrics generated from one domain generalize to another, or must they be rebuilt for each dataset?

3. Could the rubric construction be integrated online during RL training rather than pre-computed?

---

### Official Review · Reviewer_gaki · 2025-11-01

**Soundness:** 3
**Presentation:** 3
**Contribution:** 2
**Rating:** 4
**Confidence:** 3

**Summary:**

This paper addresses the critical issue of spurious reasoning in Multimodal Large Language Models (MLLMs), which often learn logically flawed shortcuts when trained with reinforcement learning based solely on final-answer correctness. To tackle this, the authors propose AutoRubric-R1V, a framework that integrates process-level supervision without relying on human annotation or stronger teacher models. The key innovation is a scalable self-aggregation method that automatically generates problem-specific rubrics by distilling consistent reasoning steps from the model's own successful solution trajectories. During RL training, an LLM-as-a-judge uses these rubrics to provide a fine-grained process reward, which is combined with the standard outcome reward. This dual-reward mechanism not only leads to state-of-the-art performance on multiple multimodal reasoning benchmarks but, crucially, also significantly enhances the model's reasoning faithfulness.

**Strengths:**

- Clear Motivation and Writing: The paper is exceptionally well-written and clearly motivated.
- Innovative and Practical Method: The core method of generating rubrics via self-aggregation is both innovative and practical. To the best of my knowledge, this is the first rubric-based method in MLLM.

**Weaknesses:**

The method is not for visual problems: A key limitation is that the text-based rubrics overlook fundamental visual perception errors. The framework can correct flawed logic in the text but fails to penalize mistakes in visual grounding (e.g., misreading a chart), limiting its effectiveness in truly multimodal scenarios.

Missing the performance on pure text tasks: The paper's evaluation is confined to multimodal tasks. Without experiments on established text-only benchmarks (like MATH500[1] or AIME24[2]), the claim of improving general reasoning faithfulness remains unsubstantiated, and the method's broader applicability is an open question.

Missing the performance on general benchmark: The method requires easily verifiable answers to work, yet its greatest value would be in complex, open-ended tasks where such verifiers don't exist (e.g., MM-Star[3], HallusionBench[4], and [5]).

[1] https://huggingface.co/datasets/HuggingFaceH4/MATH-500
[2] https://huggingface.co/datasets/math-ai/aime24
[3] Are We on the Right Way for Evaluating Large Vision-Language Models?
[4] HallusionBench: An Advanced Diagnostic Suite for Entangled Language Hallucination and Visual Illusion in Large Vision-Language Models
[5] Guided Visual Search as a Core Mechanism in Multimodal LLMs

**Questions:**

For mathematical problems, I believe final-answer verification is sufficient, making complex rubric-based methods potentially unnecessary. I hope the authors report performance directly on plain text outputs [1, 2] and standard general benchmarks [3, 4, 5] for a more direct and comprehensive evaluation.

---

### Official Review · Reviewer_Qmbm · 2025-11-01

**Soundness:** 2
**Presentation:** 3
**Contribution:** 1
**Rating:** 4
**Confidence:** 4

**Summary:**

This paper presents AutoRubric-R1V, a framework that enhances reinforcement learning in multimodal large language models (MLLMs) by providing fine-grained, rubric-based rewards. Instead of simply rewarding correct final answers as methods like RLVR do, it offers detailed, step-by-step supervision. The system autonomously creates task-specific evaluation rubrics by identifying and aggregating consistent reasoning patterns from the model's own correct solutions, requiring no manual effort or pre-existing teacher model. Tests on six multimodal reasoning benchmarks show substantial performance gains, especially in the faithfulness of the reasoning process.

**Strengths:**

1. Automated rubric creation combined with reinforcement learning enhances multimodal reasoning.
2. Experimental results across multiple multimodal knowledge and math benchmarks demonstrate consistent performance improvements.

**Weaknesses:**

1. The novelty is primarily engineering-focused, as the work constitutes an adaptation of existing methods rather than the introduction of a fundamentally new paradigm.
2. The study lacks enough robustness analysis; it does not investigate how noisy trajectory samples could adversely affect the quality of the aggregated rubric.
3. The analysis lacks depth in ablating the rubric construction process, leaving components like step comparison methods and sampling diversity unexplored.
4. The evaluation of faithfulness relies solely on GPT-4o, which is a model-based measure that lacks validation through human assessment.
5. A significant omission is the lack of failure analysis, such as instances where the rubric-based supervision led to erroneous guidance or training failures.

**Questions:**

1. How does the rubric generator handle semantic variations across different reasoning trajectories? Could mismatched wording cause false negatives?
2. How to ensure that the judge's evaluation capability is sufficient to cover the validation of all key reasoning checkpoints?
3. How sensitive is performance to the λ parameter (rubric vs. answer reward balance)?
4. Since rubrics are generated from successful model outputs, could this create a “rich-get-richer” bias?
5. Line 238 mentions: "The key intuition is that if a particular step consistently appears in many correct trajectories, it is likely to represent a causally essential component of the reasoning process; in contrast, steps that appear only sporadically are more likely to be spurious or unnecessary." However, in practice, this approach only selects general and universal steps while overlooking many problem-specific, critical, or even decisive steps. This may limit the method's applicability to only simple, general, and regular conventional problems. For such problems, couldn’t models often learn similar patterns simply through extensive training?
6. Line 248 mentions: "prompt an LLM to compare trajectories in S and summarize their common steps into an ordered set of key checkpoints." How can we ensure that the key reasoning checkpoints generated by the LLM are logically ordered, free from hallucinations, and rigorously correct?
7. Could you provide a more objective faithfulness metric (e.g., human or symbolic verification)?
8. Were more complex and strict construction and filtering mechanisms implemented, such as multi-round verification or hybrid validation?
9. Are the generated rubrics transferable across datasets or models?
10. Were cases where the definition of key reasoning checkpoints failed or the judge model malfunctioned compared and analyzed?

---

### Official Review · Reviewer_UeBw · 2025-11-04

**Soundness:** 2
**Presentation:** 2
**Contribution:** 2
**Rating:** 2
**Confidence:** 4

**Summary:**

- This paper introduces AutoRubric-R1V, a framework that integrates RLVR with process-level supervision to address spurious reasoning in Multimodal LLMs. AutoRubric-R1V integrates aggregation-based rubric generation from multiple successful trajectories generated by the model itself, without relying on external human annotation or proprietary teacher models.
- LLM-as-a-Judge is employed to perform rubric-guided evaluation, calculating a reward signal as the fraction of satisfied checkpoints in a trajectory, which is then combined with the traditional outcome reward using a weighted sum within the GRPO optimization framework.
- AutoRubric-R1V demonstrates superior performance across several multimodal reasoning benchmarks while improving reasoning faithfulness by achieving the lowest reported reasoning inconsistency rate.

**Strengths:**

- This work proposes a plausible methodology to resolve the coarse reward signal, one of the primary bottlenecks of RLVR, by integrating fine-grained process supervision.
- The paper conducts a thorough comparison against various baseline methods and provides qualitative case studies to analyze the meaningfulness of the rubric-based approach.

**Weaknesses:**

- My primary concern is the significant and seemingly downplayed computational cost. The framework requires invoking an LLM-as-a-Judge for every individual rollout during training, which appears computationally prohibitive. The paper lacks a thorough analysis of this complexity, and it is questionable whether the reported performance gains (seemingly marginal in Table 2 compared to other reasoning models) justify this substantial overhead.
- The rubric generation process has several potential loopholes. Rubrics are generated once from a warmed-up model and then held static, raising concerns about staleness as the policy model evolves. Furthermore, the paper notes only 67% rubric coverage, implying the method fails on more difficult samples where process supervision is most needed, falling back to the very vanilla RLVR it criticizes. Another important aspect is that there is no guarantee that the constructed rubrics are necessarily valid (e.g., the model might be consistently wrong), nor that the evaluations provided by LLM-as-a-judge are sufficiently reliable.
- The paper frames "rubric-based rewards" as a novel contribution, but the concept of using partial credit or key-step completion to mitigate sparse binary rewards is well-established in RLVR literature, e.g., Hint-Completion GRPO (EMNLP 2025), PS-GRPO (NeurIPS 2025), Reasoning-SQL (COLM 2025), among an extensive body of work exploiting partial credits in RLVR. The paper fails to explicitly differentiate its contribution from existing structured-reward frameworks, obfuscating the precise novelty of the work.

**Questions:**

Please refer to weaknesses section.

---

### Note · Authors · 2025-11-23

**Comment:**

Thanks for all the reviews and recommendations. We’ve taken everything seriously into consideration, and we agree that it requires further analysis and additional experiments. We will continue refining it in the future.

**Withdrawal Confirmation:**

I have read and agree with the venue's withdrawal policy on behalf of myself and my co-authors.